# α-Linolenic Acid Screened by Molecular Docking Attenuates Inflammation by Regulating Th1/Th2 Imbalance in Ovalbumin-Induced Mice of Allergic Rhinitis

**DOI:** 10.3390/molecules27185893

**Published:** 2022-09-11

**Authors:** Mengyue Ren, Yi Wang, Lin Lin, Shaoqiang Li, Qinhai Ma

**Affiliations:** 1College of Traditional Chinese Medicine, Guangdong Pharmaceutical University, Guangzhou 510006, China; 2Guangdong Second Traditional Chinese Medicine Hospital, Guangzhou 510095, China; 3State Key Laboratory of Respiratory Disease, National Clinical Research Center for Respiratory Disease, Guangzhou Institute of Respiratory Health, The First Affiliated Hospital of Guangzhou Medical University, Guangzhou 510515, China

**Keywords:** α-Linolenic acid, allergic rhinitis, molecular docking, mouse model, Th1/Th2 imbalance

## Abstract

α-Linolenic acid (ALA) is a natural essential fatty acid widely found in plant seed oils and beans, which shows positive anti-inflammatory and antiallergic effects. In our previous study, ALA was proven to bind tightly to the seven protein targets closely associated with allergic rhinitis (AR) by molecular docking, which indicates that ALA may have a potential role in the treatment of AR. A mouse model of AR induced by ovalbumin (OVA) was adopted in this study to explore the therapeutical effect and potential mechanism of ALA in treating AR. Results demonstrated that ALA remarkably relieved the nasal symptoms, reduced the OVA-sIgE level in the serum, relieved the histopathological injuries, and downregulated the mRNA expression levels of IL-6 and IL-1β in the nasal mucosa. ALA also remarkably moderated the imbalance of Th1/Th2 cells, increased the mRNA expression levels of T-bet and STAT1, and reduced GATA3 and STAT6. ALA was proven to have a substantial therapeutic effect on mice with AR, and the underlying mechanism was likely to be the regulation of Th1/Th2 imbalance through the JAK/T-bet/STAT1 and JAK/GATA3/STAT6 pathways. This study provides a specific experimental basis for the clinical use and drug development of ALA in the treatment of AR.

## 1. Introduction

Allergic rhinitis (AR) is a chronic allergic inflammation of nasal mucosa caused by allergen exposure of susceptible individuals, which is mainly mediated by immunoglobulin E (IgE) and characterized by paroxysmal sneezing, nasal obstruction, nasal itching, and rhinorrhea [1]. As one of the most common chronic diseases, AR seriously affects people’s normal life. The incidence is as high as 12–15% worldwide, and severe cases can be accompanied by pharyngeal itching, asthma, and even loss of smell [2]. Glucocorticosteroids, antihistamines, mast cell stabilizers, leukotriene receptor antagonists, and decongestants are often used in the treatment of AR in clinics; although most of these can alleviate allergic symptoms, patients are prone to recurrent attacks after the interruption of treatment, and even cause side effects, such as drowsiness, obesity, and osteoporosis after long-term use [1,3]. Therefore, finding safe and effective drugs for AR treatment with no adverse effects is of great importance.

α-Linolenic acid (ALA) is an essential ω-3 polyunsaturated fatty acid widely found in plant seed oil and must be supplied via diet [4]. ALA has the physiological functions of preventing atherosclerosis, cardiovascular, and cerebrovascular diseases, regulating blood lipids, reducing obesity, and shows positive anti-inflammatory and antiallergic effects [5,6,7,8].

Molecular docking investigation, which has been widely used in drug design and discovery [9,10], was performed in our previous study to predict the potential role of ALA in the treatment of AR [11]. Seven protein targets, namely, 5-lipoxygenase (5-LO), histamine H1 receptor (HRH1), corticosteroid-binding globulin (CBG), M1 muscarinic acetylcholine receptor (mAChR M1), M3 muscarinic acetylcholine receptor (mAChR M3), phosphodiesterase 4B (PDE4B), and prostaglandin D2 (PGD2), closely associated with AR, were collected to dock with ALA by using Sybyl-7.3 software. As shown in Table 1 and Figure 1, the molecular docking scores are all greater than 5, ranging from 5.8 to 9.7, indicating that ALA may bind well to the seven protein targets, and the most stable molecular conformation was visualized on Pymol 2.4.0 (Schrödinger, NY, USA). Although no studies have been reported, ALA may have a potential role in the treatment of AR.

The imbalance in type 1 helper T (Th1) cells and type 2 helper T (Th2) cells has been considered to be the main induction factor in AR [12,13]. In this study, a mouse model of AR induced by ovalbumin (OVA) was adopted to explore the therapeutic effect and the possible mechanism, which could provide a scientific basis for the further study of ALA in the treatment of AR.

## 2. Results

### 2.1. ALA Ameliorated Nasal Symptoms in OVA-induced AR in Mice

The nasal symptoms of mice in each group were evaluated immediately after the last intranasal challenge. As shown in Figure 2A,B, the numbers of sneezing and rubbing actions of the mice in the AR model group (54.4 and 27.6 times) were significantly increased compared with that in the control group (4.1 and 4.3 times, *p* < 0.01; *p* < 0.01). The sneezing and rubbing actions of mice were remarkably relieved after the oral administration of ALA. Compared with the model group, the numbers of sneezing and rubbing were significantly reduced to 35.2 and 24.0 times (*p* < 0.01 and *p* < 0.05) in the ALA-L group and 22.0 and 15.9 times (*p* < 0.01 and *p* < 0.05) in the ALA-H group.

### 2.2. ALA Reduced the OVA-sIgE Level in the Serum

The OVA-sIgE level in the serum of mice was detected by using an enzyme-linked immunosorbent assay (ELISA); IgE production has been regarded as an important indicator of the occurrence of AR [2]. As shown in Figure 2C, the OVA-sIgE level in the model group was significantly increased compared with that in the control group (*p* < 0.01). The oral administration of a low dose of ALA (ALA-L) and a high dose of ALA (ALA-H) significantly reduced the OVA-sIgE level in a dose-dependent manner (*p* < 0.01 and *p* < 0.01). No significant difference was observed between the ALA-H and the dexamethasone (Dex) groups.

### 2.3. ALA Relieved the Histopathological Injuries in the Nasal Mucosa

To observe the effects of ALA on inflammatory damages and goblet cell proliferation in the nasal mucosa of mice, histological analysis was performed by hematoxylin and eosin (HE) and periodic acid-schiff (PAS) staining. The HE staining results (Figure 2D) showed that the epithelial cells of nasal mucosa were disarranged; moreover, part of the cilium disappeared, and obvious eosinophil infiltration and interstitial edema occurred in the model group. After the oral administration of ALA, the nasal mucosa injuries were substantially relieved. In the ALA-H and Dex groups, after the epidermal cells of nasal mucosa were arranged in order, most of cilia appeared to be intact, and fewer eosinophils were observed. The PAS staining results (Figure 2E) showed that the numbers of goblet cells in the nasal mucosa of the model group was remarkably higher compared with that of the control group. The number of goblet cells was substantially reduced after treatment with ALA.

### 2.4. Effect of ALA on the mRNA Expression Levels of IL-6, IL-1β, IFN-γ, and IL-4 in the Nasal Mucosa

To determine whether ALA suppressed the release of inflammatory mediators and regulate the Th1/Th2 imbalance in AR mice, the mRNA expression levels of IL-6, IL-1β, IFN-γ (mainly released by Th1 cells), and IL-4 (mainly released by Th2 cells) in the nasal mucosa were detected by quantitative reverse transcription-polymerase chain reaction (qRT-PCR). As depicted in Figure 3A,B, the mRNA expression levels of IL-6 and IL-1β in the model group were significantly increased compared with that in the control group (*p* < 0.01 and *p* < 0.01). The oral administration of ALA at doses of 500 and 2000 mg/kg significantly reduced the expression levels of IL-6 and IL-1β in a dose-dependent manner (*p* < 0.01, *p* < 0.01, *p* < 0.01, and *p* < 0.01), which demonstrated that ALA can inhibit the release of inflammatory mediators in the nasal mucosa of AR mice. As shown in Figure 3C,D, the mRNA expression of IFN-γ in the model group was significantly decreased (*p* < 0.01), and that of IL-4 was significantly increased (*p* < 0.01) compared with that in the control group. The treatment of ALA at doses of 500 and 2000 mg/kg significantly enhanced the expression of IFN-γ in the nasal mucosa (*p* < 0.01 and *p* < 0.01), whereas it distinctly reduced the expression of IL-4 (*p* < 0.01 and *p* < 0.01), indicating that ALA can regulate the Th1/Th2 imbalance in AR mice effectively.

### 2.5. Effect of ALA on the Percentages of CD3^+^CD4^+^IFN-γ^+^ Th1 and CD3^+^CD4^+^IL-4^+^ Th2 Cells in the Spleen

To evaluate the effect of ALA on the ratio of CD3^+^CD4^+^IFN-γ^+^ Th1 and CD3^+^CD4^+^IL-4^+^ Th2 cells in the spleen mononuclear cells (SMCs) of AR mice, the percentages of Th1 and Th2 cells were analyzed and measured via flow cytometry. As shown in Figure 4, the percentages of Th1 cells in the model group (1.24%) was significantly lower than that in the control group (3.04%, *p* < 0.01), whereas the percentages of Th2 cells in the model group was 5.1%—significantly higher than the 1.69% in the control group (*p* < 0.01). The percentages of Th1 cells in the ALA-L and ALA-H groups were significantly increased to 1.83% and 2.09% (*p* < 0.05 and *p* < 0.01), and the percentages of Th2 cells in the ALA-L and ALA-H groups were significantly decreased to 2.83% and 1.52% (*p* < 0.01 and *p* < 0.01) after the oral administration of ALA. No obvious difference was observed between the ALA-H and Dex groups.

### 2.6. Effect of ALA on the mRNA Expression Levels of T-bet, GATA3, STAT1, and STAT6 in the Nasal Mucosa

To explore whether ALA inhibited the allergic inflammatory responses through the JAK/STAT signal pathway, the mRNA expression levels of T-bet, GATA3, STAT1, and STAT6 in the nasal mucosa were detected by qRT-PCR. As shown in Figure 5, the mRNA expression levels of T-bet and STAT1 in the model group were significantly decreased (*p* < 0.05 and *p* < 0.01), whereas that of GATA3 and STAT6 were significantly increased (*p* < 0.01 and *p* < 0.01) compared with that in the control group. Compared with the model group, the oral administration of ALA at doses of 500 and 2000 mg/kg significantly enhanced the expression levels of T-bet (*p* < 0.01 and *p* < 0.05) and STAT1 (*p* < 0.01 and *p* < 0.01), whereas it significantly reduced the mRNA expression levels of GATA3 (*p* < 0.01 and *p* < 0.01) and STAT6 (*p* < 0.05 and *p* < 0.05). The oral administration of ALA at 2000 mg/kg dosage was more effective than Dex in reducing the expression of GATA3 and increasing the expression of STAT6.

## 3. Discussion

ALA is a natural essential fatty acid widely found in plant seed oils and beans, which can be metabolized into two important long chain omega-3 fatty acids in the body, docosahexaenoic acid (DHA) and eicosapentaenoic acid (EPA) [5]. Similar to its derivatives DHA and EPA, ALA plays an important role in preventing cardiovascular and cerebrovascular diseases [14,15], lowering blood lipids [16], is anticancer [17], and also shows obvious physiological anti-inflammatory and antiallergic activities [7,18,19]. Therefore, ALA was believed to have a potential role in treating AR, an allergic inflammatory disease, and was predicted by molecular docking technology, which has long been recognized as a key method with substantial effect in drug discovery and design [20].

In our previous study, seven protein targets, namely, 5-LO, HRH1, CBG, mAChR M1, mAChR M3, PDE4B, and PGD2, closely associated with AR, were collected to dock with ALA. 5-LO is a key enzyme that catalyzes the formation of leukotrienes from arachidonic acid, which plays an extremely important role in the occurrence and development of AR [21]. As the histamine receptor most closely related to allergic inflammatory response, HRH1 can promote the release of histamine and other inflammatory factors, and stimulate the activity of B cells in AR [22]. CBG is one of the effective targets for the treatment of AR, and drugs that can be used to prevent and treat AR and asthma, such as beclomethasone, have been designed and developed through this target [23]. The mAChR M1 and M3 are abundantly expressed in the nasal mucosa glands and can induce mucus hypersecretion after activation by acetylcholine; moreover, mAChR M3 plays a role in regulating AR nasal mucosal vasodilation [24]. PDE4 inhibitors are found to exhibit potent anti-inflammatory effects in vivo and in vitro, and they can be used to treat inflammatory airway diseases effectively such as AR [25,26]. PGD2S plays an important role in the synthesis of prostaglandin D2, which is mainly produced by the degranulation of activated mast cells after allergen exposure and performs antigen cross-linking with the high-affinity receptors of IgE [27]. The high docking scores between ALA and the seven protein targets above indicated that ALA may have a potential role in the treatment of AR.

In this research, a mice model of AR induced by OVA was adopted to evaluate the therapeutical effect and mechanism of ALA. The results showed that ALA remarkably reduced the numbers of sneezing and rubbing actions, reduced the OVA-sIgE level in the serum, relieved the histopathological injuries and goblet cell proliferation in the nasal mucosa, and down-regulated the mRNA expression levels of IL-6 and IL-1β. The production of IgE is an important indicator of the occurrence of AR, and the inflammatory cytokines, such as IL-6 and IL-1β, are released by mast cells and eosinophils when the IgE antibodies bind to specific high-affinity receptors on the surface of these inflammatory cells, which subsequently triggers rhinitis symptoms [28]. Apart from eosinophil infiltration and interstitial edema, goblet cell hyperplasia is an important pathological feature in the nasal mucosa with AR, which indicates that more mucus is secreted [29]. Thus, the pharmacodynamic results show that ALA has a substantial effect on the treatment of AR.

The imbalance in Th1 and Th2 cells has been considered as the main induction factor in AR [12,13]. Under normal conditions, CD4^+^ T cells (Th0 cells) will differentiate into Thl and Th2 cells in a certain proportion, and they exist in a relatively balanced state. Abnormal immune responses are produced, and the balance of Th1/Th2 is broken when the body is exposed to allergens that trigger AR [30,31]. In this study, the mRNA expression of IFN-γ (mainly released by Th1 cells) and the percentages of CD3^+^CD4^+^IFN-γ^+^ Th1 cells in the AR model group were substantially decreased. The IL-4 (mainly released by Th2 cells) and the percentages of CD3^+^CD4^+^IL-4^+^ Th2 cells were substantially increased compared with that in the control group, demonstrating that the imbalance of Th1/Th2 was clearly induced in the AR mice. The imbalance of Th1/Th2 occurred in nasal mucosa and SMCs were substantially restored after the oral administration of ALA, indicating that ALA may play a therapeutic role in AR by regulating Th1/Th2 imbalance.

To further explore the underlying mechanism of ALA in regulating Th1/Th2 imbalance in AR, the JAK/STAT signaling pathway, which plays an important role in Th1 and Th2 immune responses [32,33], has been investigated subsequently. When the secreted IFN-γ binds to its receptor, the T-bet gene will be activated and expressed through the JAK/STAT1 signal transduction pathway, and the expression of T-bet will enhance the differentiation of Th0 cells to Th1 cells [34]. When the Th2 cytokine IL-4 binds to its receptor, STAT6 will be activated by the expression of phosphorylated JAK1/JAK3, thereby activating the transcription factor GATA3, which can promote the differentiation of Th0 cells into Th2 cells and produce Th2 cytokines such as IL-4, IL-5, and IL-13 [35]. In this study, ALA increased the mRNA expression levels of T-bet and STAT1 and reduced the mRNA expression levels of GATA3 and STAT6 in AR mice, demonstrating that ALA may attenuate the inflammation in AR mice by regulating the imbalance of Th1/Th2 through the JAK/T-bet/STAT1 and JAK/GATA3/STAT6 pathways.

In conclusion, ALA was proven to have a substantial therapeutic effect on AR in a mouse model induced by OVA for the first time. The main underlying mechanism of ALA in attenuating inflammation is likely to be the regulation of Th1/Th2 imbalance through the JAK/T-bet/STAT1 and JAK/GATA3/STAT6 pathways. This study provides a certain experimental and theoretical basis for the clinical use and the drug development of ALA in the treatment of AR.

## 4. Materials and Methods

### 4.1. Reagents

Dexamethasone (purity ≥ 98%) and OVA were purchased from Sigma Chemical Co. (St. Louis, MO, USA). Aluminum hydroxide (AL(OH)_3_) was purchased from Damao Chemical Reagent Factory (Tianjin, China). ALA (purity ≥ 90%) was purchased from Shanxi Longzhou Biological Technology Co. (Shanxi, China). TRIzol reagent was purchased from Invitrogen (Carlsbad, CA, USA). Red blood cell lysis buffer was obtained from Solarbio (Beijing, China). Mouse OVA specific IgE ELISA kit was obtained from BioLegend (San Diego, CA, USA). RPMI 1640 and fetal bovine serum (FBS) were obtained from Gibco (Grand Island, NY, USA). Prime Script RT reagent kit and TB Green Premix Ex Taq II were obtained from TaKaRa (Osaka, Japan). Leukocyte activation cocktail, purified rat anti-mouse CD16/CD32 (mouse BD Fc block) antibody, and BV421-labeled anti-mouse CD3 antibody were purchased from BD (Franklin Lakes, NJ, USA). Fluorescein isothiocyanate (FITC)-labeled anti-mouse CD4 antibody, phycoerythrin (PE)-labeled anti-mouse IFN-γ antibody, allophycocyanin (APC)-labeled anti-mouse IL-4 antibody, and intracellular fixation and permeabilization buffer set kit were purchased from eBioscience (San Diego, CA, USA).

### 4.2. Animals

Specific pathogen-free (SPF) male BALB/c mice (18–22 g) were obtained from the Experimental Animal Center of Southern Medical University (Certificate number SCXK 2016-0041), and the experiment was approved by the Undergraduate Laboratory Animal Center of Guangdong Pharmaceutical University (Approval number gdpulac2019200). The mice were fed adaptively for a week before the experiment, and were free to take standard food and water under SPF conditions at 20–24 °C and with a relative humidity of 50–60%.

### 4.3. Ovalbumin-Induced AR Model and ALA Treatment

The AR mouse model was achieved as summarized in Figure 6. Briefly, on days 0, 7, and 14, the mice were intraperitoneally injected with 200 μL of suspended sensitizer (0.5 mg OVA and 20 mg AL(OH)_3_ to every milliliter of saline, adjusted the pH to the physiological pH with) as base sensitization. On days 21–34, 20 μL of 4% OVA solution was instilled in each nasal cavity once daily for challenge with a laboratory pipette. Fifty mice were randomly divided into five groups: the control, the AR model, the Dex, the ALA-L, and the ALA-H groups. Except for the control group, the mice in other groups needed to be modeled by OVA as described above. On days 21–34, the mice in the Dex, ALA-L, and ALA-H groups were administered once daily with 2.5 mg/kg of Dex and 500 and 2000 mg/kg of ALA by intragastric administration (0.2 mL/20 g), respectively. The mice in the control and AR model groups were given distilled water. The experiment lasted for 35 days, and the intranasal challenge was initiated 30 min after each gavage.

### 4.4. Evaluation of Nasal Symptoms

The numbers of sneezing and nasal scratching of mice in each group were counted for 30 min immediately after the last intranasal challenge with 4% OVA. On day 35, the mice were sacrificed, and blood samples were collected and centrifuged (4000 rpm for 10 min) to obtain the serum. Spleen samples were aseptically removed and stored in cold saline for the cell isolation of SMCs, and nasal mucosa samples were removed for histopathological and qRT-PCR analyses.

### 4.5. Histological Analysis

Nasal mucosa samples were fixed in 4% paraformaldehyde for 24 h and were then embedded in paraffin. Embedded samples were prepared into sections with a 4 µm thickness and stained with HE and PAS. Histopathological changes were evaluated and photographed by using a semi-electric fluorescence microscope (Olympus, Tokyo, Japan).

### 4.6. Measurement of OVA-sIgE in the Serum by ELISA Assay

The level of OVA-sIgE was measured following the manufacturer’s instructions in the mouse OVA-sIgE ELISA kit. The reaction was stopped by adding the stop solution, and the absorbance was read within 10 min at 450 nm by using a microplate spectrophotometer (Multiskan Sky, ThermoFisher, Waltham, MA, USA).

### 4.7. Isolation of SMCs

The removed spleen was crushed aseptically on a 200-mesh filter by using a sterile syringe handle, and the cells were evenly dispersed in 2 mL of phosphate-buffered saline (PBS) in a test tube. Red blood cell lysis buffer (6 mL) was added to the tube, mixed by inversion, placed on ice for 15 min, and centrifuged at 2000 rpm for 10 min at 4 °C. After being lysed twice, the cells were washed twice in 5 mL of PBS, washed once in 5 mL of 1640 medium, centrifuged at 2000 rpm for 10 min, resuspended with RPMI 1640 medium containing 10% FBS and adjusted to 4 × 10^6^/mL.

### 4.8. Flow Cytometry Analysis

The obtained SMCs were seeded on a 24-well plate and stimulated with the leukocyte activation cocktail (containing phorbol 12-myristate 13-acetate, ionomycin, and brefeldin A) for 5 h in a 37 °C humidified CO2 incubator. Subsequently, the cells were distributed into tubes, washed once with PBS, and incubated with an Fc block at room temperature for 15 min to avoid the non-specific antibody binding via Fc receptors in antibody labeling. For surface staining, the cells were stained with BV421-labeled anti-mouse CD3 and FITC-labeled anti-mouse CD4 antibodies for 30 min at room temperature away from light. Subsequently, the cells were fixed and permeabilized with intracellular fixation and permeabilization buffer, and then incubated with PE-labeled anti-mouse IFN-γ and APC-labeled anti-mouse IL-4 at room temperature for 15 min away from light. The stained cells were washed and resuspended in PBS, and detected by using an FACSCelesta flow cytometer (BD, Franklin Lakes, NJ, USA).

### 4.9. RNA Isolation and qRT-PCR Analyses

The nasal mucosa samples in each group were homogenized in 1 mL of TRIzol at 4 °C, and the total RNA was isolated following the manufacturer’s instruction. Subsequently, genomic DNA was removed, and complementary DNA (cDNA) was obtained via reverse transcription reaction using a Prime Script RT reagent Kit. Following the manufacturer’s instruction of TB Green Premix Ex Taq II kit, PCR mix was prepared on ice and qRT-PCR was performed for 40 cycles (95 °C for 5 s and 60 °C for 30 s) in a CFX96 Touch Real-Time PCR detection system (Bio-Rad, Berkeley, CA, USA). The sequences of the IL-1β, IL-6, IFN-γ, IL-4, T-bet, GATA-3, STAT-1, STAT-6, and GAPDH are listed in Table 2. The expression levels of the eight target genes were normalized relative to GAPDH by using cycle threshold method.

### 4.10. Statistical Analysis

All results were expressed as mean ± standard deviation (SD) and analyzed on IBM SPSS Statistics 20 software (Chicago, IL, USA). One-way ANOVA coupled with a post-hoc test (LSD *t*-test or Tamhane’s T2 test) was used to compare the differences in the groups, and *p* < 0.05 indicated statistical significance.

## Figures and Tables

**Figure 1 molecules-27-05893-f001:**
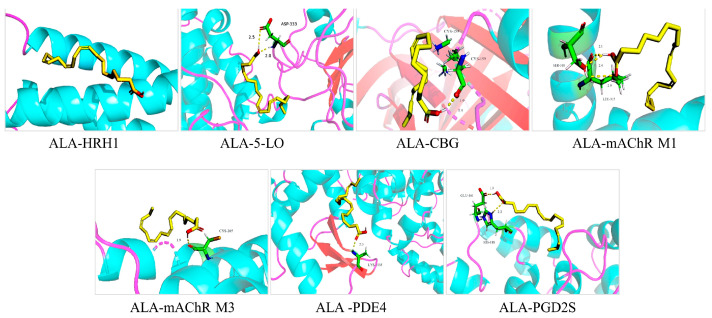
Molecular docking diagram of ALA with 5-LO, HRH1, CBG, mAChR M1, mAChR M3, PDE4 B, and PGD2. 5-LO, 5-lipoxygenase; HRH1, histamine H1 receptor; CBG, corticosteroid-binding globulin; mAChR M1, M1 muscarinic acetylcholine receptor; mAChR M3, M3 muscarinic acetylcholine receptor; PDE4B, phosphodiesterase 4B; PGD2, prostaglandin D2.

**Figure 2 molecules-27-05893-f002:**
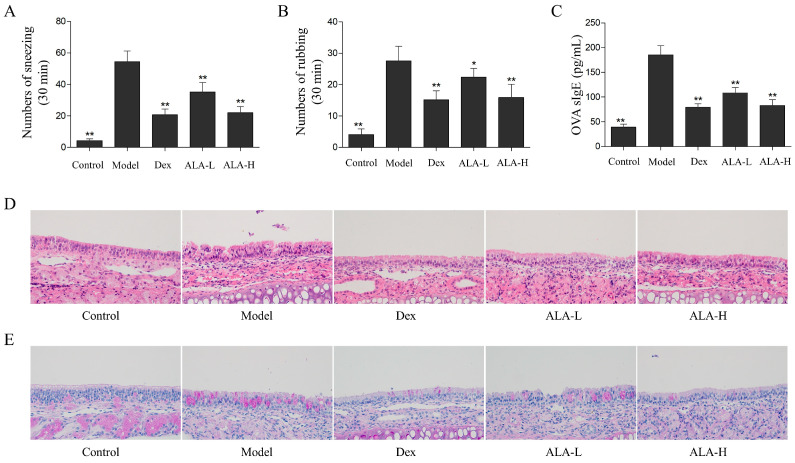
Effect of ALA on OVA-induced AR in mice. The numbers of sneezing (**A**) and rubbing actions (**B**) of mice were counted for 30 min immediately after the last intranasal challenge. (**C**) The level of OVA-sIgE in serum were measured by enzyme-linked immunosorbent assay (ELISA). The HE (**D**) and PAS (**E**) staining were used to observe the histopathological changes in nasal mucosa samples of mice. Between-group comparisons were performed using one-way ANOVA coupled with LSD *t*-test (equal variances assumed) or Tamhane’s T2 test (equal variances not assumed). Data are expressed as mean ± SD; *n* = 10; * *p* < 0.05, ** *p* < 0.01 versus AR model. Dex, dexamethasone; ALA-L, low dose of α-linolenic acid; ALA-H, high dose of α-linolenic acid.

**Figure 3 molecules-27-05893-f003:**
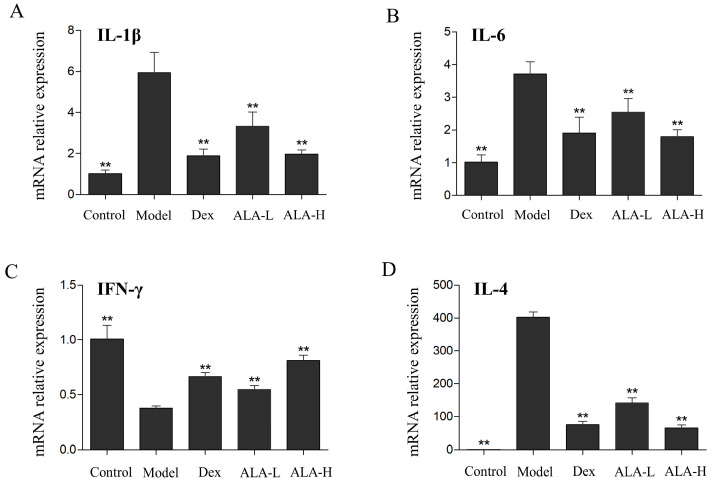
Effect of ALA on the mRNA expression levels of IL-6 (**A**), IL-1β (**B**), IFN-γ (**C**), and IL-4 (**D**) in the nasal mucosa of AR mice. Between-group comparisons were performed using one-way ANOVA coupled with LSD *t*-test (equal variances assumed) or Tamhane’s T2 test (equal variances not assumed). Data are expressed as mean ± SD; *n* = 4; ** *p* < 0.01 versus AR model. Dex, dexamethasone; ALA-L, low dose of α-linolenic acid; ALA-H, high dose of α-linolenic acid.

**Figure 4 molecules-27-05893-f004:**
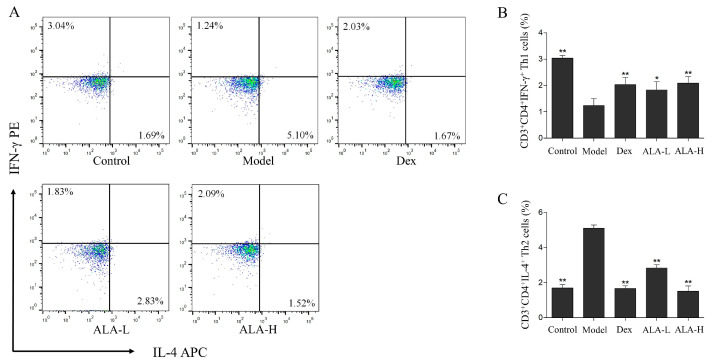
Effect of ALA on the percentages of CD3^+^CD4^+^IFN-γ^+^ Th1 and CD3^+^CD4^+^IL-4^+^ Th2 cells in the spleen of AR mice. Representative dot plots of CD3^+^CD4^+^IFN-γ^+^ Th1 (in the upper left quadrant) and CD3^+^CD4^+^IL-4^+^ Th2 cells (in the lower right quadrant) (**A**), and the percentages of CD3^+^CD4^+^IFN-γ^+^ Th1 cells (**B**) and Th2 cells (**C**) in the SMCs separated from the spleen samples are analyzed via flow cytometry. Between-group comparisons were performed using one-way ANOVA coupled with LSD *t*-test (equal variances assumed) or Tamhane’s T2 test (equal variances not assumed). Data are expressed as mean ± SD; *n* = 3; * *p* < 0.05, ** *p* < 0.01 versus AR model. Dex, dexamethasone; ALA-L, low dose of α-linolenic acid; ALA-H, high dose of α-linolenic acid.

**Figure 5 molecules-27-05893-f005:**
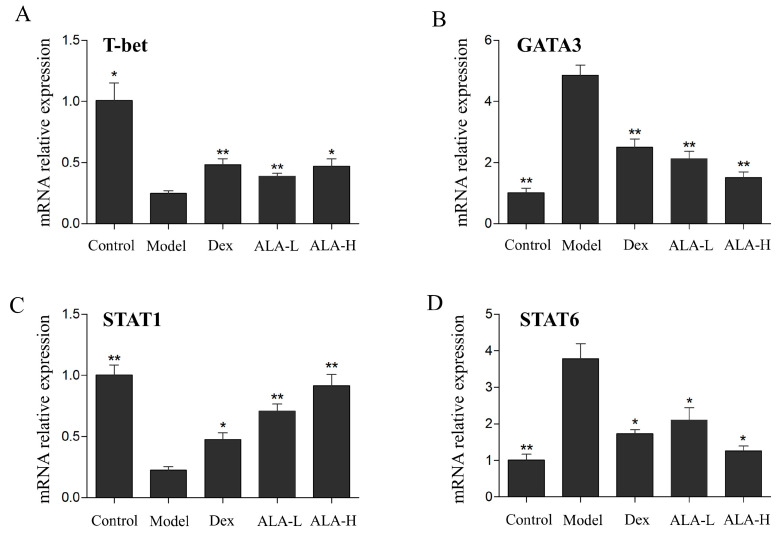
Effect of ALA on the mRNA expression levels of T-bet (**A**), GATA3 (**B**), STAT1 (**C**), and STAT6 (**D**) in the nasal mucosa in AR mice. Between-group comparisons were performed using one-way ANOVA coupled with LSD *t*-test (equal variances assumed) or Tamhane’s T2 test (equal variances not assumed). Data are expressed as mean ± SD; *n* = 4; * *p* < 0.05, ** *p* < 0.01 versus AR model. Dex, dexamethasone; ALA-L, low dose of α-linolenic acid; ALA-H, high dose of α-linolenic acid.

**Figure 6 molecules-27-05893-f006:**
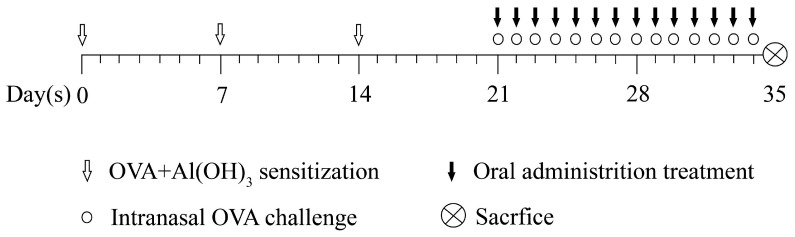
Flow chart of the OVA-induced AR model and ALA administration.

**Table 1 molecules-27-05893-t001:** Docking scores and binding energy of ALA with 5-LO, HRH1, CBG, mAChR M1, mAChR M3, PDE4B, and PGD2.

Targets	PDB ID	Docking Scores [11]	Binding Energy (kJ/mol)
HRH1	3RZE	5.77	−6.5
5-LO	3V98	8.85	−9.1
CBG	4C41	9.11	−11.8
mAChR M1	6ZFZ	8.28	−12.2
mAChR M3	4DAJ	9.66	−6.9
PDE4B	2QYL	8.72	−9.5
PGD2S	7M8W	7.52	−7.7

Note: 5-LO, 5-lipoxygenase; HRH1, histamine H1 receptor; CBG, corticosteroid-binding globulin; mAChR M1, M1 muscarinic acetylcholine receptor; mAChR M3, M3 muscarinic acetylcholine receptor; PDE4B, phosphodiesterase 4B; PGD2, prostaglandin D2.

**Table 2 molecules-27-05893-t002:** Primer sequences for qRT-PCR.

Gene	Primer	Sequences (5′-3′)
IL-1β	Forward	GCAACTGTTCCTGAACTCAACT
	Reverse	ATCTTTTGGGGTCCGTCAACT
IL-6	Forward	TAGTCCTTCCTACCCCAATTTCC
	Reverse	TTGGTCCTTAGCCACTCCTTC
INF-γ	Forward	TCAAGTGGCATAGATGTGGAAGAA
	Reverse	TGGCTCTGCAGGATTTTCATG
IL-4	Forward	ACAGGAGAAGGGACGCCAT
	Reverse	GAAGCCCTACAGACGAGCTCA
T-bet	Forward	AGCAAGGACGGCGAATGTT
	Reverse	GTGGACATATAAGCGGTTCCC
GATA-3	Forward	AAGCTCAGTATCCGCTGACG
	Reverse	GTTTCCGTAGTAGGACGGGAC
STAT1	Forward	TCACAGTGGTTCGAGCTTCAG
	Reverse	CGAGACATCATAGGCAGCGTG
STAT6	Forward	CTCTGTGGGGCCTAATTTCCA
	Reverse	CATCTGAACCGACCAGGAACT
GAPDH	Forward	ACCTGCCAAGTATGATGACATCA
	Reverse	GGTCCTCAGTGTAGCCCAAGAT

## Data Availability

Data is contained within the article.

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
