# Peer review of "α-Linolenic Acid Screened by Molecular Docking Attenuates Inflammation by Regulating Th1/Th2 Imbalance in Ovalbumin-Induced Mice of Allergic Rhinitis"

_molecules, 2022, doi:10.3390/molecules27185893_

Round 1
Reviewer 1 Report
The idea of the research work is interesting, however there are lots of language mistakes and grammatical errors through the whole manuscript.
The firstly mentioned briefs should include the full name and the brief.
I have attached the PDF of the manuscript including some comments that could help to improve the manuscript.
Methods: The methods part must be improved. The experiments methods should be mentioned in details to make them repeatable for other researchers especially the AR model. The questions added to the methods part in the attached manuscript should be answered to clarifiy how the method was performed.
Results: The figures are of bad resolution. They all need to be improved, they are not placed correctly in the correct places for them as they are shifted to the left. All the briefs in the figures should be placed in the footenote to make the figure self explanatory.
Why did the authors use different number of animals in each experiment (sometimes 10 , 3 or 4 )?
The affinity values and the data of the docking results must be added.
The disscusion part could be improved.

Author Response
Thank you very much for careful reading our manuscript and for giving valuable comments and suggestions. We have revised it according your recommendations and revised portions are marked in red in the paper. We hope the revised portion could meet the publication requirements.
1. Comment: There are lots of language mistakes and grammatical errors through the whole manuscript.
Response:The language mistakes and grammatical errors in the manuscript have been modified by the professional people and some inaccurate concepts have been also modified in the revised version.
2. Comment: The firstly mentioned briefs should include the full name and the brief.
Response:The full names and the brief have been listed in the revised manuscript when appeared for the first time, such as ALA-L, ALA-H, Dex, HE, PAS, and ELISA.
3. Comment: The methods part must be improved. The experiments methods should be mentioned in details to make them repeatable for other researchers especially the AR model. The questions added to the methods part in the attached manuscript should be answered to clarify how the method was performed.
(1) Question 1: Why did the mice drink deionized water?
Response:Thank you for reminding this typo in our original version. The mice were free to take standard food and water provided by the Undergraduate Laboratory Animal Center of Guangdong Pharmaceutical University, the word “deionized” has been removed.
(2) Question 2: Was any buffer used to adjust the pH to the physiological pH?
Response:The AR model was established according to the method described by Feng S et al (Feng, S., et al. Therapeutic Effect of C-C Chemokine Receptor Type 1 (CCR1) Antagonist BX471 on Allergic Rhinitis. Journal of Inflammation Research, 2020, 13: 343-356). In our study, when the sensitizer solution (0.5 mg OVA and 20 mg AL(OH)3 to every milliliter of saline) for intraperitoneal injection was prepared, the pH of the solution increases slightly, and citric acid was used to adjust the pH to the physiological pH before injection. In the revised version, the operation of adjusting the pH of the sensitizer solution has been added in corresponding position in the “4.3 Ovalbumin-Induced AR Model and ALA Treatment”.
(3) Question 3: how was the oil suspension prepared for ip injection?
Response:In our study, like dexamethasone, oily ALA was administered to mice by intragastric administration, and only the sensitizer solution (0.5 mg OVA and 20 mg AL(OH)3 to every milliliter of saline) was prepared for intraperitoneal injection.
(4) Question 4: How was the 4% OVA solution instilled into the mice noses?
Response:In our study, the 4% OVA solution for challenge was instilled into the mice noses with a laboratory pipette (10-100 μL), which is accurate in quantification, and very simple and easy to use to instill into the mouse nose. The using of laboratory pipette in establishing AR model has been added in the revised manuscript.
(5) Question 5: How the dexamethasone, ALA-L, and ALA-H were orally administered to the mice?
Response:In our study, the dexamethasone, ALA-L, and ALA-H solution were prepared with distilled water and administered to mice by intragastric administration (0.2 mL/20 g) once a day for 14 days. In the revised version, “orally” have been modified to the more accurate descriptions in the “4.3 Ovalbumin-Induced AR Model and ALA Treatment”.
(6) Question 6: 2000 should be the high dose?
Response:Thank you for reminding this typo in our original version. In the “4.3 Ovalbumin-Induced AR Model and ALA Treatment”, “2000 and 500 mg/kg of ALA” has been modified to “500 and 2000 mg/kg of ALA”.
(7) Question 7: How was the counting performed for each animal per group?
Response:In our study, the numbers of sneezes and nasal rubbing movements were counted by six blinded observers who were unaware of the sample’s identity, and one observer observes the performance of five mice at a time. Since most of the sneezes and nose scratching behaviors occurred within the first ten minutes of the nasal instillation, each mouse was given a 10-minute nasal drip interval to facilitate observation.
4. Comment: The figures are of bad resolution. They all need to be improved, they are not placed correctly in the correct places for them as they are shifted to the left. All the briefs in the figures should be placed in the footenote to make the figure self explanatory.
Response:Thank you for your valuable comments. For better observation, the higher-resolution pictures of all the figures have been replaced, and placed in the corresponding correct places in the revised version. Besides, all the briefs in the figures have been placed in the footnote to make the figures self-explanatory, such as “Dex, dexamethasone”, “ALA-L, low dose of α-linolenic acid”, “ALA-H, high dose of α-linolenic acid”.
5. Comment: Why did the authors use different number of animals in each experiment (sometimes 10, 3 or 4)?
Response: In our study, there were 10 mice in each group. The numbers of sneezing and nasal scratching of all the mice were counted and analyzed before they were sacrificed, therefore, the number of samples is 10 in the evaluation of nasal symptoms. At the same time, since the IgE production has been regarded as an important indicator of the occurrence of AR, the OVA-sIgE level in the serum of all the mice was also detected. In addition, the nasal mucosa samples of mice were required for both qRT-PCR and histopathology experiments. The mouse nasal mucosa is very small, and it is difficult to judge whether the nasal mucosa is damaged or not when it is removed. Therefore, the number of samples for qRT-PCR and histopathology experiments were set at 4 and 6, respectively, to ensure that a structurally intact section of the nasal mucosa can be obtained in histological analysis. Besides, in our original design of this study, the spleens of mice might also be used for protein or genetic testing, so in order to keep the sample size for these experiments, the number of samples in the flow cytometry experiment was only 3, which can still be statistically analyzed.
6. Comment: The affinity values and the data of the docking results must be added.
Response:Thank you for your valuable comments. A Table 1 entitled “Docking scores and binding energy of ALA with 5-LO, HRH1, CBG, mAChR M1, mAChR M3, PDE4 B, and PGD2” have been added in the revised version, and the docking scores and binding energy of ALA with the seven protein targets that closely associated with AR have been added. Meantime, “Table 1. Primer sequences for qRT-PCR” in original manuscript has been modified to “Table 2. Primer sequences for qRT-PCR” in corresponding position.
7. Comment: The discussion part could be improved.
Response:Thank you for your advice. In the “3. Discussion” part, the language errors and grammatical errors have been modified, and the contents have been improved in the revised manuscript.
In addition, according to the suggestions made by other reviewers, we also have specified the detailed statistical methods in figure captions and in “4.10. Statistical Analysis” in the revised manuscript.
Reviewer 2 Report
It is not clear whether the in silico data reported in figure 1 have been previously published. Please specify in figure caption.
Regarding the statistical approach, authors should specify whether they applied a post hoc test coupled to ANOVA. Please specify in figure captions and in statistical analysis paragraph.
The quality of figures should be improved.
Author Response
Dear reviewer, thank you very much for careful reading our manuscript and for giving valuable comments and suggestions. We have revised it according your recommendations and revised portions are marked in red in the paper.
1. Comment: It is not clear whether the in silico data reported in figure 1 have been previously published. Please specify in figure caption.
Response:Thank you for your valuable suggestions. According to the suggestions made by other reviewers, A Table 1 entitled “Docking scores and binding energy of ALA with 5-LO, HRH1, CBG, mAChR M1, mAChR M3, PDE4 B, and PGD2” have been added in the revised manuscript. The data of molecular docking scores between ALA and the seven protein targets that closely associated with AR have been published in our previous study, and the reference have been marked in Table 1. In addition to this, no other data or images in Table 1 and Figure 1 have been published publicly.
2. Comment: Regarding the statistical approach, authors should specify whether they applied a post hoc test coupled to ANOVA. Please specify in figure captions and in statistical analysis paragraph.
Response:Thank you for your valuable suggestions. In our study, a post-hoc test has been applied to couple to ANOVA in the statistical approach, and LSD t-test (when equal variances are assumed) and Tamhane’s T2 test (nonparametric test; when equal variances are not assumed) were chosen in the post-hoc test. In the revised manuscript, the detailed statistical methods have been specified in figure captions and in “4.10. Statistical Analysis”.
3. Comment: The quality of figures should be improved.
Response:For better observation, the higher-resolution pictures of all the figures have been replaced, and placed in the corresponding correct places in the revised version. Besides, all the briefs in the figures have been placed in the footnote to make the figures self-explanatory, such as “Dex, dexamethasone”, “ALA-L, low dose of α-linolenic acid”, “ALA-H, high dose of α-linolenic acid”.
In addition, according to the suggestions made by other reviewers, we also have made modification as follows:
(1) The language mistakes and grammatical errors in the manuscript have been modified by the professional people and some inaccurate concepts have been also modified in the revised version.
(2) “Table 1. Primer sequences for qRT-PCR” in original manuscript has been modified to “Table 2. Primer sequences for qRT-PCR” in corresponding position.
(3) The full names and the brief have been listed in the revised manuscript when appeared for the first time, such as ALA-L, ALA-H, Dex, HE, PAS, and ELISA.
(4) The operation of adjusting the pH of the sensitizer solution has been added in corresponding position in the “4.3 Ovalbumin-Induced AR Model and ALA Treatment”.
Round 2
Reviewer 1 Report
the manuscript is much improved. The required modifications have been done except for the figures resolution. The figures are still so hazy and unclear due to the low quality. You can add the original figures from their source as this affects the quality of results presentation.